# Social Environment and Attitudes toward COVID-19 Anti-Contagious Measures: An Explorative Study from Italy

**DOI:** 10.3390/ijerph20043621

**Published:** 2023-02-17

**Authors:** Alessandro Gennaro, Matteo Reho, Tiziana Marinaci, Barbara Cordella, Marco Castiglioni, Cristina Liviana Caldiroli, Claudia Venuleo

**Affiliations:** 1Department of Dynamic Clinical and Health Psychology, Sapienza University of Rome, 00185 Rome, Italy; 2Department of Human and Social Sciences, Salento University, 73100 Lecce, Italy; 3Department of Human Sciences “R. Massa”, University of Milano-Bicocca, 20126 Milan, Italy

**Keywords:** pandemic, adherence, satisfaction, culture, sense-making

## Abstract

Social and cultural aspects (i.e., political decision making, discourses in the public sphere, and people’s mindsets) played a crucial role in the ways people responded to the COVID-19 pandemic. Framed with the Semiotic-Cultural Psychological Theory (SCPT), the present work aims to explore how individual ways of making sense of their social environment affected individuals’ perception of government measures aimed at managing the pandemic and the adherence to such measures. An online survey was administered from January to April 2021 to the Italian population. Retrieved questionnaires (N = 378) were analyzed through a Multiple Correspondence Analysis (MCA) to detect the factorial dimensions underpinning (dis)similarities in the respondents’ ways of interpreting their social environment. Extracted factors were interpreted as markers of Latent Dimensions of Sense (LDSs) organizing respondents’ worldviews. Finally, three regression models tested the role of LDSs in supporting the individual satisfaction with the measures adopted to contain the social contagion defined at national level, individual adherence to the containment measures and the perception of the population’s adherence to them. Results highlight that all the three measures are associated with a negative view of the social environment characterized by a lack of confidence in public institutions (health system, government), public roles and other people. Findings are discussed on the one hand to shed light on the role of deep-rooted cultural views in defining personal evaluations of government measures and adherence capacity. On the other hand, we suggest that taking into account people’s meaning-making can guide public health officials and policy makers to comprehend what favors or hinders adaptive responses to emergencies or social crises.

## 1. Introduction

The global COVID-19 crisis has resulted in governments taking very strict measures to contain the spreading of the virus (i.e., social distancing, self-isolation, increased hygiene procedures and lockdown) forcing people to re-shape their relationships and daily activities. Research evidence studied citizens’ attitudes, compliance and adherence toward the restrictive measures as key factors in counteracting the pandemic [1,2]. Several works evidenced that adherence (or not) to containment measures depends on individual characteristics such as gender [3,4], marital status [5], occupation [6], geographical area [7], income [8], the presence of family and friends [9], the political environment and the timing of each government’s introduction of measures [10]. Such results were consistent even in the Italian context; for example, Lorettu and colleagues [11] highlighted adherence to be associated with gender, education, geographical area, occupation and age. Similarly, Carlucci and colleagues [12] found that women, the most educated people, residents of Southern Italy, middle-aged people and health workers were more willing to adhere to the quarantine guidelines. However, such evidence is conflicting: for example, Galende and colleagues [13] found that younger people were less willing to adhere, while Daoust [14] states that older people, while being the most at-risk population group, do not always seem to be among the most reactive in terms of adherence to constraint measures. Taken together, the obtained results suggest that individual characteristics seem not be reliable benchmarks for detecting individual predispositions to adhere to pandemic contrast measures; nor do they offer an intervention perspective able to increase adherence or compliance.

Literature focusing on psychological factors such as attitudes [15], sense of control [16] and risk perception [17,18] suggests that adherence to pandemic contrast measures depends on their perceived efficacy [3,19,20,21,22,23], individual perceptions of stress [24], individual coping strategies [25,26], perceived self-efficacy [27], individual health beliefs [28], and perceived susceptibility to the virus [29]. Moreover, Ciacchella and colleagues [30] found that nonadherence was predicted by higher levels of dissociation, which in turn predicted low levels of sense of community and high levels of hopelessness. Such results were confirmed even in the Italian context. Carlucci and colleagues [12] found that people with high levels of risk perception, anxiety and sensitivity to the risk of contracting the virus had higher levels of adherence to restrictive measures. Similarly, Nese and colleagues [31] reported that, concerning the restrictive measures, women showed higher levels of anxiety, intolerance of uncertainty and perceived risk than men.

Further studies have focused on the psycho-social factors associated with adherence, highlighting how the latter is also influenced by social norms [32,33], social support [34], civic engagement [35,36] and trust in government [37,38], which seems to be a central factor influencing adherence to containment measures [3,10,39,40]. However, Wong and Jensen [41] pointed out that this can lead to a paradox; in fact, according to the authors, trust can lead people to underestimate the risks associated with the spread of the virus and thus reduce individual willingness to adhere to containment measures. Moreover, trust in government seems to be influenced also by cultural factors such as individualism and collectivism. For example, Travaglino and Moon [42] found that vertical collectivism predicted greater shame, while horizontal collectivism predicted stronger confidence in government action.

To the best of our knowledge, only three studies focused the role of sense-making and cultural dimensions in affecting people’s adherence to restrictive measures: in the Italian context, Procentese and colleagues [43] investigated sense-making processes in relation to the pandemic, showing that the restrictive measures made by the Italian government were interpreted in terms of adaptation to the health emergency. This interpretation of the restrictions involved further attribution of new meanings to, for example, meaningful relationships, where such redefinition resulted, according to the authors, in an influence on individuals’ emotional, cognitive and activation arousal, as well as on socially embedded sense-making processes. Similarly, Milman and colleagues [44,45] evidenced that stressful or traumatic events—in this case, the COVID-19 pandemic—caused mental suffering by disrupting meaning and creating processes that violate individual’s core beliefs, namely implicit constructs regarding the world as a fair place where we individuals can influence our circumstances [46].

Finally, Nair and colleagues [47] pointed out that different cultural dimensions, such as individualism and collectivism, power distance, uncertainty avoidance, masculinity and femininity, and future orientation, differentiate among protective behaviors adopted by different countries during the pandemic, and propose that cultural awareness should become an important element of crisis response policy. Despite the low amount of research work, several psychological theories recognize that attitudes and the representation of phenomena in the social environment are affected by the interpretation of the social context in which they occur. For example, some studies have shown how social context perceptions based on a foe-friend scheme may reflect the need to cope with the uncertainty of the context of belonging [48,49]. Similarly, from a psychoanalytic perspective, it has been suggested that affect-laden interpretations drive the meaning attributed to discrete elements of experience [50,51]. Again, Weiss and Cropanzano [52] showed that offenders’ acts are interpreted in negative or positive terms on the basis of wider representations such as in-group versus out-group belonging or dispositional versus situational causes.

Framed in this line of thought, the present work aims to shed light on the role of an individual’s way of making sense of a social environment in affecting the individual’s perception of government measures aimed at managing the pandemic and the adherence to such measures. Specifically, according to the Semiotic-Cultural Psychological Theory (SCPT) [53,54,55], we explored the role of affective dimension grounding individuals’ worldviews in predicting people’s adherence and satisfaction with the COVID-19 containment measures adopted by the Italian government.

### 1.1. Theoretical Framework

The SCPT [53,54,55] posits that people’s sense-making is grounded in basic affect-laden latent dimensions (LDSs) having a bipolar structure (e.g., pleasant versus unpleasant, passivity versus engagement) embedded in sociocultural contexts (see, for instance, [46,56,57,58,59,60,61], which organize the sense of self and the world (e.g., the belief that the world is a nice place to live or that it is an inherent threat) and drive individuals’ social experience (see [51,62]).

Previous studies highlighted the role of LDSs in organizing individuals’ worldviews which lay down the way to feel, act and face social phenomena: for example, Veltri and colleagues [63] analyzed the role played by LDSs in the “leave/remain” preference at the Brexit referendum and showed that regions of the United Kingdom with a high incidence of “leave” votes could be fully differentiated from those with a low incidence of “leave” votes according to LDSs. Mannarini and colleagues [64] analyzed the role played by LDSs in the voting behavior of a representative sample of the Italian electorate and found that anomic feelings and/or identity motivations favored voting for populist parties through their negative impact on political attitudes. Venuleo and colleagues [65] point out that the LDSs by which students interpret their role and educational context influence the likelihood of dropping out of higher education. Again, Cordella and colleagues [66] estimated the impact of LDSs on COVID-19 vaccine compliance on a representative sample of the Italian population and found that trust in institutions was associated with vaccine compliance and that this relationship was moderated by LDSs.

The acknowledgement that people’s ways of interpreting and acting on their social experience are guided by affect-laden assumptions implies recognizing that in the context of the COVID-19 pandemic as well the public adherence towards advice and recommendations does not depend simply on the clarity of the information offered (i.e., individual and social risks related, numbers of deaths due to the contagion). Advice and recommendations are not legitimated by the observation of the events, nor by logical and analytical thinking [67,68]; rather they are considered on the basis of their consistency with beliefs, feelings, and actions underpinned by the LDS.

For example, a person whose LDSs could be interpreted in terms of a “community of fate” would interpret the compliance with the government’s measures to be mandatory in order to fight together with the community the war against the health emergency. Opposite LDSs, in contrast, support the view, as “masters of one’s own life”, that takes any limitation, including measures and rules, as a violation of one’s own freedom. Social and political processes (i.e., media, scientists, health and economic policies) influence how individuals make sense of their outer and inner realities and their capacity to produce interpretations aligned to the affect-laden assumptions. For instance, the uncertainty of the social-cultural scenario can compromise the capacity to produce more fine-grained and differentiated evaluations (see [69,70]). According to the SPCT, thus, we do not conceive LDSs as intra-psychic structures; people interpret their identity and social experience on the basis of the meanings available in their context of belonging. The LDSs are patterns of meanings that are active within the social group’s cultural milieu [71,72]. This field has to be conceived of as the semiotic field that frames similarities as well as differences among members’ sense-making (and related values, statements, attitudes and behavior).

Framing individuals’ perceptions of enacted measures to manage the pandemic and the adherence to them into the SCPT background would help to recognize how reactions to events deal with wider attitudes toward life scenarios in which people are involved and, most of all, would offer cues to bridge individual meaning making and enacted behaviors.

### 1.2. Aims

The above-mentioned SCPT perspective suggests the importance of considering ordinary people’s perspectives to understand the ways that citizens experience, evaluate, cope, and react to adopted government measures to contain the health emergency. Accordingly, the present study aims to explore the role of LDSs, grounding individuals’ worldviews, in predicting people’s adherence and satisfaction with the COVID-19 containment measures adopted by the Italian government. Such a perspective, whether not directly focused on the COVID-19 scenario, is not new in literature. Previous studies suggest that protective and prosocial behaviors are related to a positive view of the social environment (e.g., a sense of trust toward institutions and other people, a sense of belonging and social connectedness) [73,74,75,76,77,78,79]. Research work on disaster situations shows that the capacity to go beyond the absolutization of one’s worldview [69] is crucial for collective and proactive responses [80]. In contrast, a negative view of the social environment—e.g., feeling of living in a anomic and untrustful context where you cannot rely on anyone and/or institutions are perceived as untrustful—was found to be related to risky behaviors [65,81,82,83,84], as well as—in the context of the COVID-19 scenario—to conspiracy theories and negationist discourses and attitudes about the nature of the virus [85]. Thus, based on the existing studies, we expect that a low level of compliance, poor expectations towards the level of compliance expressed by other people, and low satisfaction can be interpreted as some of the ways the subject acts out the view of the relationship with the sphere of collective life (e.g., institutions, rules, other people) in terms of disappointment, conflict and distrust.

## 2. Materials and Methods

### 2.1. Participants

An online survey was distributed in the Italian context through social networks to explore the role of LDSs in supporting or undermining individual adherence and satisfaction with the measures adopted by the institutions to contain the COVID-19 pandemic. In response to the first pandemic wave, several interventions were deployed by the Italian government to contain the infection, firstly applied to the so-called “red-zone” (Lombardia and fourteen provinces of Veneto, Emilia Romagna, Piemonte and Marche), then to the whole country (Decree of the President of the Council of Ministers, 9 March 2020). School closure was established at the national level on 5 March 2020 and a national lockdown (stay-home mandate and closure of all nonessential productive activities) issued on 11 March, then eased after 4 May 2020 [86]. During the national lockdown, people were asked to keep a distance from others and to avoid any form of public and social gatherings, thus, to stop visiting relatives and friends, to stop praying in the churches, to stop doing sports in the gym and in the park, to stop visiting museums, watching films at the cinema, assisting in social and cultural events, as well as to close their commercial activities, offices, cafes and shops. The second wave of COVID-19 began in Italy in September 2020 and then sharply increased, pushing the government to establish at the beginning of November a system of physical distancing measures organized in progressively restrictive tiers imposed on a regional basis according to epidemiological risk assessments.

We collected data from January 2021 till April 2021. The choice of the time window to gather data reflects the need to gather individuals’ perceptions about the anti-contagious measures on one hand, avoiding the moment of maximum emergency in which the perception of their usefulness and adherence to them could be more or less taken for granted given the serious situation on the Italian territory (for example, during the 2020 lockdown), and on the other hand to gather data in a short-term perspective in which the restriction measures, although less pervasive, had entered people’s daily habits.

A sample of 378 questionnaires was collected (mean age: 35.87, SD: 14.143; male 29.3%, females: 71.7%). Participation was voluntary and according to the aim of the work, namely, to explore Italian perceptions of anti-contagious measures. The only inclusion criteria were being adult (>18 years), living in the Italian territory and being able to understand the Italian language.

### 2.2. Measures

The View of Context questionnaire (VOC) [87] was used to map the LDS through which people interpret their social context. The VOC is a self-reporting instrument composed of 29 items aimed at exploring the respondents’ perceptions, judgments and opinions about micro/macro social environment (e.g., evaluation of the place where the respondents live, level of reliability of social services, such as police, hospitals, schools and so on) and moral/social values (e.g., study, stay with family, respect and so on). It is worth noting that the items are designed to trigger generalized meanings—rather than, for instance, to prompt circumstantiated reasoning. To this end, the items concern generic objects (e.g., ‘Italian people’, ‘Italy’), which are more likely to work as—as indeed they were—a projective stimulus. Furthermore, items are associated with response modes that force the respondent to position himself in relation to a contrasting position. This makes the structure of the response isomorphic to the oppositional structure of the LDSs. Each item is associated with a four-point Likert scale ranging from 1 = totally disagree to 4 = totally agree (for details on the methodology, see [88]). VOC proved to have satisfactory construct validity and internal consistency (α = 0.70) [87].

Adherence to and satisfaction with the containment measures were assessed according to three specific items that were used. Respondents were asked to rate on a five-point Likert scale (1 = not at all; 2 = a little; 3 = quite 4 = completely; 5 = I don’t know) (a) individual satisfaction, (b) individual adherence and (c) their perception of the population’s adherence to the containment measures defined at the national level. The issues of the Likert scales were defined according to a consensus procedure between four researchers having at least Ph.D. research experience.

All procedures performed in the study complied with the ethical standards of the institutional research committee and with the 1964 Helsinki declaration and its later amendments or comparable ethical standards. Participants were informed about the general aim of the research, the anonymity of responses, and the voluntary nature of participation and signed informed consent. No incentive was given. The project was approved by the Ethics Commission for Research in Psychology of the Department of History, Society and Human Studies of the University of Salento (protocol n. 16881 of 2021/01/28).

### 2.3. Data Analysis

The method of analysis applied to the responses to the VOC is grounded on the assumption that the meanings are shaped in terms of response variability. Multiple Correspondence Analysis (MCA) [89] is consistent with this principle. MCA has been applied through SPAD v5.5 software (Decisia, Pantin, France). MCA allows us to define LDS by summing up the relations observed among nominal data by using a limited number of factorial dimensions. Each factorial dimension extracted describes the juxtaposition of two patterns of strongly associated (co-occurring) response modes (i.e., the tendency to respond with the modality m to the item A, the modality n to the item B, the modality u to the item C, and so forth). Insofar as the co-occurring responses (m, n, u…) have no reciprocal semantic linkage, their aggregation lends itself to being interpreted as the effect of a LDS linking the response modalities independently of their specific content [90].

We focused on the first two factorial dimensions (henceforth: VOC 1 and VOC 2) extracted from each MCA, as the ones explaining a relevant proportion of the data matrix’s inertia. We adopted the subjects’ scores (factorial coordinates) on the two factorial dimensions as measures of their LDSs. The higher the respondent’s factorial coordinate, the higher the degree of association between the respondent’s profile of answers and the profile characterizing one of the two polarities of the factor/dimension of meaning.

The answers to the three items investigating compliance and satisfaction with the measures adopted at the national level have been transformed into a quasi-cardinal scale following the Thorgerson method which, according to the Thurstone approach, suggests that each interviewee’s response is based on a latent variable having normal distribution [91,92]. Accordingly, for each modality of each variable Xj, we calculated the absolute frequency. In our case, five modalities (corresponding to the five points of the Likert scale) for each of the three variables were considered. Then we computed the cumulative relative frequency, representing the estimation of the cumulative density function Fj(i) of the normal distribution. Finally, the Φ−1[Fj(i)] represents the inverse function of the normal standard distribution to compute quantile, τj, of the function.

Finally, to test the role of LDSs in explaining respondents’ satisfaction with and adherence to the government’s containment measures, the two factorial coordinates measuring the LDSs (i.e., VOC 1 and VOC 2) and the quantified evaluation concerning individuals’ adherence to and satisfaction with COVID-19 containment measures have been transformed into z points and subjected to a regression analysis. Specifically, three different regression models, having each measure rating as independent variables and VOC 1 and VOC 2 as dependent variables were performed through the SPSS software version 26.0 (IBM, Armonk, NY, USA).

## 3. Results

The MCA applied to the answers to the VOC questionnaire detected two factorial dimensions. The matrix under analysis being highly dispersive, the variance was re-evaluated through the Benzécri [93] formula. The first factor (VOC 1) explained the 47.394% of the re-evaluated variance and the second factor (VOC 2) explained the 21.193%. Table 1 and Table 2 illustrate, respectively, the VOC 1 and VOC 2 reporting the ten more representative (positive and negative) items and their level of association (V-test) with the factor [94]. Moreover, in Figure 1 we reported individuals’ distribution in the retrieved factors’ plan.

Retrieved factors have been interpreted according to the items characterizing them. On one hand, VOC 1 has been interpreted in terms of trust in the social environment: social trust (−) versus social untrust (+). As a matter of fact, such a dimension opposes two patterns of answers that have been interpreted as the markers of LDS consisting of two ways of evaluating the social environment. Social trust (−). The pattern of answer modalities highlights trust in social structures such as the government and the public administration, as well as trust in people (for instance, disagreement is expressed with respect to statements such as “it is useless to turn to those in public roles” or “nowadays people do not know who to rely on”) and the future (“it will be better”). Within this scenario, a feeling of having a role in influencing what happens is at stake (for instance, strong disagreement is expressed in statements such as “it is useless to bargain, so it is not possible to influence what happens” and “my life is controlled largely by the case”).

Social untrust (+). The pattern of answers highlights a lack of confidence in public institutions (health system, government), public roles (high agreement is expressed with respect to statements like “it is useless to turn to those in public roles”) and other people (“nowadays people do not know who to rely on”). Within this critical scenario, the future appears to be uncertain (“It is not possible to predict the future”), and both the sense of individual agency and rules appear to be compromised (“it is useless to bargain; you cannot influence what happens so much”, “nowadays, people are forced to live day by day, “in life it’s important not to have too many scruples”).

On the other hand, VOC 2 has been interpreted in terms of modes of acting in the social environment: Moderation (−) versus Reactivity (+). This dimension opposes two patterns of answers that have been interpreted as the markers of LDS consisting of two different models of acting in the social environment. Differently from VOC 1, VOC 2 lends itself to be interpreted not in terms of its content, but in terms of the response modality, and as such is reflected in the tendency to modulate judgments and positions or to generalize the connotations of the objects.

Moderation (−). Answers adopting intermediate choices on the Likert scale (“quite agree” or “quite disagree”) are aggregated, marking a generalized modality of perceiving experience characterized by an attitude of moderation, such as reflected in the tendency to modulate judgment and positions toward the object of the experience. The contents of the answers reflect on the one hand the sense of individual agency (enough agreement is expressed in statements such as “my life is determined by my actions” and enough disagreement about statements such as “my life is under the control of influential people” and “my life is controlled largely by chance”); on the other hand, the sense of living in a critical scenario which is not totally under individual control (“Nowadays people are forced to live day by day”; “It is not possible to make predictions about the future”).

Reactivity (+). Answers adopting the extreme choices on the Likert scale (“Strongly agree” or “strongly disagree”) are aggregated, reflecting a reactive attitude towards experience. In this case, the sense of individual agency (“My life is determined by my actions”) and the feeling of living in a very strong critical scenario where you cannot rely on other people seems to be interpreted in a self-referential way. In other words, the respect for the rules can be subordinated to the interests of ones’ own group of belonging (“sometimes you need to break the rules to help loved ones”) and a day-by-day attitude toward experience (“Nowadays people are forced to live day by day”).

A chi square tested the distribution of respondents according to gender into the quadrants of the retrieved factors plan. No differences related to gender were found (χ^2^ 2.346; df 3, *p* = 0.504).

The regression models highlighted meaningful results. Specifically, the satisfaction with measures adopted to contain the social contagion is meaningfully predicted by both VOC 1 and VOC 2, namely trust (VOC 1) and the modes of acting in the social environment (F = 29.933; *p* < 0.001) with a R^2^ of 0.138. The regression model concerning the personal level of adherence to the contagion containment measures was significantly (F = 4.448; *p* = 0.012) with a R^2^ of 0.023 predicted by the modes of acting in the social environment. Moreover, modes of acting in the social environment predicted meaningfully the perception of the level of adhesion of the general population containment measures (F = 5.619; *p* = 0.004) with a R^2^ of 0.029.

The parameters of the regression models reported in Table 3 highlight the specificity of the effects. Specifically, a significant effect of both the factorial dimensions (VOC 1: β = −0.333; *p* = 0.001; VOC 2: β = −0.164; *p* = 0.001) was found on the satisfaction with the measures adopted to contain the COVID-19 contagion. The negative scores indicate that the social trust and the moderation poles (respectively, the polarity indicated by the negative scores of VOC 1 and VOC 2) are associated with higher levels of satisfaction (Item 1).

A significant effect of VOC 1 was found either on the personal level of adherence to the contagion containment measures (item 2) (β = −0.128; *p* = 0.012), and on the perceived level of the general population’s adherence to the containment measures (β = −0.171; *p* = 0.001) with the negative scores indicating an association between social trust and personal (Item 2) and general adherence (Item 3).

## 4. Discussion

According to the SPCT, we argued that the individuals do not deal with single events or contents of experience, including norms and measures adopted by governments to contain the COVID-19 pandemic emergency, but that rather they react to the single event as a function of their more general way of interpreting the scenario where they live. Consistently, the current study explores the way in which individuals’ attitudes toward the government measures (adherence and satisfaction) were predicted by LDS grounding people’s views of the world.

It is worth noting that the first retrieved factorial dimension (VOC 1) has been interpreted as a global connotation of the domain of experience in positive versus negative terms, while the second one (VOC 2) concerns opposite attitudes of response related to the way of acting in the social environment: moderation versus reactivity. This result supports the way the LDSs have been conceptualized within the SPCT framework: generalized meanings homogenizing the specific objects of experience beyond their semantic peculiarity and differences. Furthermore, it is worth noting that the factorial dimensions identified in this study are consistent with the ones identified in previous studies analyzing the view of the social environment in social groups differently characterized for age, health status, and attitudes towards risk behaviors (e.g., [65,83]). Such a result supports the idea that variations of the LDSs move within the meanings active within the cultural milieu [95]: in other words, what changes is their distribution within the social group, rather than their content and reciprocal connection. The results of the regression models support our hypothesis that respondents expressing a different view of the social environment give a different assessment of the individual satisfaction, individual adherence and perceived adherence by the general population toward the measures adopted by the Italian government to contain the spread of COVID-19.

Specifically, one out of two LDSs defining the cultural field—the one labelled as trust in the social environment (VOC 1)—is associated with all the three items related to satisfaction, individual and perceived general adherence to COVID-19 containment measures. Specifically, the higher the subjects’ score on “Social untrust”, which reveals a tendency to homogenously evaluate unreliable social structures and people, the lower the satisfaction, the individual adherence and the perceived adherence by the general population. This result is easily understandable. The sense of trust in social norms and institutions and hope in the future encourages the commitment to social “feed and bond” behavior and practices consistent with social norms. In contrast, social untrust entails a “day-by-day” perspective which has no hope of a better future; one might suppose that without hope in the subject’s power to get ahead, commonly held values appear meaningless and lose their normative power [96]. The low adherence and satisfaction with the government measures can be understood as a way to remark on their disappointment in an “unhealthy” and unwelcoming world. Similarly, the perceived low adherence to the measures expressed by other people may be understood as the reflex of the affect-laden idea that we live in a world where people care only about their own interests.

Such findings are consistent with previous research emphasizing the role of social trust in enhancing protective behaviors and commitment to the community’s values and norms. In the same vein, if social trust erodes, people will be less likely to identify with the rules of their own conduct [97,98] and more likely to adopt risky behaviors [99,100]. For instance, in a study of Venuleo and colleagues [84], subjects differing in their evaluation of the macro-social environment as a reliable or unreliable place also differ in their evaluation of the degree of social approval towards illicit hazardous behavior. The more reliable the environment is perceived to be, the greater the effect on social approval of illicit behavior and the lower the effect on illicit hazardous behaviors. In other words, the more people identify with their social environment, the greater the likelihood of social disapproval towards behavior that is not legitimated by the rules and is less compatible with what society expects. Similarly, a greater concern about anti-contagious measures may reflect the common trust in official criteria on what has to be approved or not.

The satisfaction expressed toward the anti-contagious measures related also to the acting attitude in the social environment (VOC 2). The higher the subjects’ score on the reactivity pole, the lower the satisfaction. In contrast, the higher the subjects’ score on the moderation pole, the higher the satisfaction. One can see that reactivity corresponds to a condition of intense affective activation that triggers a homogenizing form of thinking and polarized evaluations [51,101]. Respondents with higher scores on this polarity express homogenous negative connotations of what is other by itself: they cannot trust anybody. This view in turn seems to feed a day-by-day attitude toward experience, as if the present was the only representable time, and a self-referential way to interpret his/her power to determinate his/her own life resting merely on personal preference and the interests of ones’ own group of belonging. Within this interpretative frame, it is understandable that measures which foreground the public interests are devalued as constraints on individual “freedom to act as one pleases”. Such results are consistent with the findings from recent studies among European societies, revealing that about 40% of the respondents view the external world as something capable of disrupting their living space [54].

## 5. Conclusions

Some limits of the present study need to be highlighted. First, this work is based on a convenience sample located in a specific geographic area, thus the content and strength of the relationship between LDSs and measures rating satisfaction and adherence might differ in other countries. What appears to be generalizable is the relationship between people’s worldviews and attitudes toward anti-contagious measures; nevertheless, the content, the strength and the nature of this relationship is probably context-specific. Second, the study adopted three single items to assess satisfaction and adherence among the respondents which could be in some way misinterpreted by respondents. Furthermore, given the nature of the analysis, the interpretations proposed above on the association between LDSs and rating on the measures cannot but be speculative. Our suggestion is that LDSs serve as a frame for driving people towards a certain evaluation of the measures. However, regression models do not allow us to take the results as evidence of a causal effect of LDSs, nor to rule out that other individual or social factors may mediate the effect of LDSs. Further research work will try to overcome these limits and should seek to establish longitudinal research designs to study how LDSs tends to modify in time. Nevertheless, the findings discussed above deserve attention, both at the theoretical and methodological level. At the theoretical level, results highlight the role of sense-making and encourage the hypothesis that people’s worldviews orient the way people interpret and therefore deal with the measures that institutions adopt to in a response to a health crisis

At the methodological level, our results suggest that a teaching top-down approach on what is needed to face a national and global crisis is not enough to solicit citizens’ compliance. Government advice traditionally took the form of transmission of knowledge about the risks and harmful implications of not respecting the measures adopted. This kind of intervention is based on the assumption that knowledge is the means by which healthy behaviors can be promoted and harmful ones prevented. However, if—as suggested by the results of this study—the way people assess the anti-contagious measure is related to their view of the social context, no preventive action can be carried out effectively without taking into account the meaning, for the target population, of acting adaptively within their social environment. Their risk evaluations are more than a mere result of an error or of a gap (of information, skill and so on), but rather a powerful semiotic organizer of the mind, orienting the way they feel and think about their experience and how they make sense of what the experts define as risky behavior (e.g., not staying at home, not wearing a mask, not avoiding social aggregation) and relate to the contents of the strategies addressed to them.

In our view, social untrust and reactivity cannot be addressed if they are understood simply as undesirable individual attitudes. As we suggested, people interpret their experience and the social environment they are part of on the basis of a defined system of meanings negotiated and shared in that context and in the frame of the semiotic resources (e.g., values, beliefs, metaphors, norms, linguistic and action codes) available in their cultural milieu. This means that communicative practices and discourses exchanged in the interpersonal contexts (e.g., family, educational and work environment) play a crucial role in shaping contents and the direction of people’s sense-making. However, the interpersonal sphere represents only one side of the issue. The feeling of living in an unreliable context where you can rely only on yourself emerged as an important constraint to the identification with measures and decisions adopted by the government; a constraint that requires innovative and effective efforts at community and political levels. The SPCT, as well as findings of studies based on the Terror Management Theory [102,103], suggest that the more the uncertainty of the scenario, the more sense-makers are likely to restore the stability of their sense-making through their adherence to generalized worldviews [104,105]. Unemployment, worsening of living conditions, lack of resources and poor political investment in sectors felt as crucial (as example health and education) feed uncertainty in the future. Consequently, the lack of hope in a better future, norms lose their regulative power as well as the desire to be part of the same community, in which individuals plays a crucial part.

## Figures and Tables

**Figure 1 ijerph-20-03621-f001:**
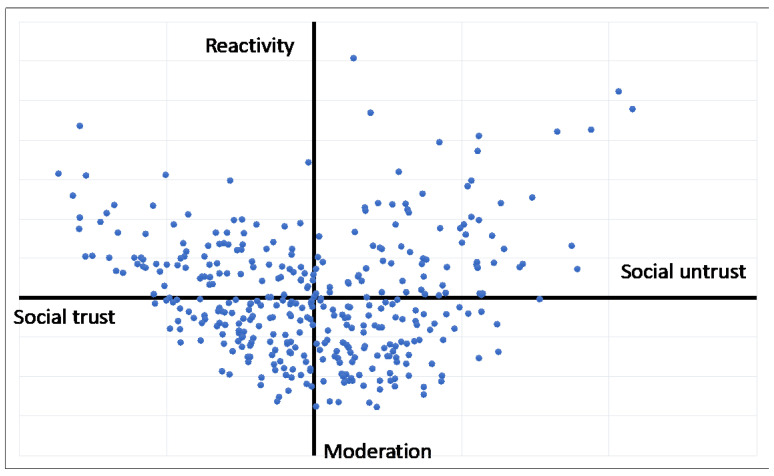
Individual positioning on the retrieved factors space.

**Table 1 ijerph-20-03621-t001:** Response modalities associated with the first factorial dimension (VOC 1).

**(** **−** **) Social Trust**		
**Item**	**Response Mode**	**V-Test**
Government	Somewhat reliable	−9.18
It is useless to bustle about, since you cannot affect what will happen	Strongly disagree	−8.92
Public administration	Somewhat reliable	−7.9
To a great extent, my life is controlled by accidental happenings	Strongly disagree	−7.81
There’s little use in writing to public officials because often they aren’t really interested in the problems of the average man	Somewhat disagree	−7.7
These days a person doesn’t really know whom he can count on	Somewhat disagree	−7.28
How will you live here?	Better	−7.22
People are unable to change	Strongly disagree	−7.05
It is not possible at all to make any provision for the future	Somewhat disagree	−7.03
Having few scruples	Not at all	−6.9
**(+) Social Untrust**		
**Item**	**Response Mode**	**V-Test**
In spite of what some people say, the lot of the average man is getting worse, not better	Strongly agree	6.58
Having few scruples	Very	6.75
There’s little use in writing to public officials because often they aren’t really interested in the problems of the average man	Strongly agree	6.81
It is not possible at all to make any provision about the future	Strongly agree	7
It is useless to bustle about, since you cannot affect what will happen	Strongly agree	7.03
It is not possible at all to make any provision for the future	Strongly agree	7.05
Health care services	Not at all reliable	7.43
These days a person doesn’t really know whom he can count on	Strongly agree	7.49
Nowadays a person has to live pretty much for today and let tomorrow take care of itself	Strongly agree	8.52
Government	Not at all reliable	9.14

**Table 2 ijerph-20-03621-t002:** Response modalities associated to the first factorial dimension (VOC 1).

**(−) Moderation**		
**Item**	**Response Mode**	**V-Test**
My life is determined by my own actions	Somewhat agree	−8.4
Nowadays a person has to live pretty much for today and let tomorrow take care of itself	Somewhat agree	−8.7
These days a person doesn’t really know whom he can count on	Somewhat agree	−7.51
In spite of what some people say, the lot of the average man is getting worse, not better	Somewhat agree	−7.4
It is useless to bustle about, since you cannot affect what will happen	Somewhat disagree	−7.05
Police	Somewhat reliable	−6.6
It’s hardly fair to bring children into the world, the way things look for the future	Somewhat disagree	−6.41
My life is chiefly controlled by powerful others	Somewhat disagree	−6.27
To a great extent, my life is controlled by accidental happenings	Somewhat disagree	−6.16
It is not possible at all to make any provision about the future	Somewhat agree	−6.12
**(+) Reactivity**		
**Item**	**Response Mode**	**V-Test**
There’s little use in writing to public officials because often they aren’t really interested in the problems of the average man	Strongly disagree	6.57
Those who succeed in life have luck on their side	Strongly disagree	6.62
My life is determined by my own actions	Strongly agree	6.72
It is useless to bustle about, since you cannot affect what will happen	Strongly disagree	7.03
Sometimes one has to break the rules to help one’s loved ones	Strongly agree	7.06
In spite of what some people say, the lot of the average man is getting worse, not better	Strongly agree	7.81
To a great extent, my life is controlled by accidental happenings	Strongly disagree	7.81
These days a person doesn’t really know whom he can count on	Strongly agree	8.17
People are unable to change	Strongly disagree	8.22
Nowadays a person has to live pretty much for today and let tomorrow take care of itself	Strongly agree	8.42

**Table 3 ijerph-20-03621-t003:** Regression models parameters.

Item	Factorial Dimensions (LDSs)	B	Std. Error	Beta	T	Sig
Satisfaction (item 1)	VOC 1 (social trust vs. social untrust)	−0.333	0.48	−0.333	−6.942	0.001
	VOC 2 (moderation vs. reactivity)	−0.164	0.48	−0.164	−3.418	0.001
Personal adherence (item 2)	VOC 1 (social trust vs. social untrust)	−0.128	0.051	−0.128	−2.512	0.012
	VOC 2 (moderation vs. reactivity)	−0.082	0.051	−0.082	−1.609	0.108
Population adherence (item 3)	VOC 1 (social trust vs. social untrust)	−0.171	0.051	−0.171	−3.351	0.001
	VOC 2 (moderation vs. reactivity)	0.006	0.051	0.006	0.121	0.905

## Data Availability

The data presented in this study are available on request from the corresponding author.

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
