# Peer review of "Social Environment and Attitudes toward COVID-19 Anti-Contagious Measures: An Explorative Study from Italy"

_ijerph, 2023, doi:10.3390/ijerph20043621_

Round 1

Reviewer 1 Report

Comments and suggestions!

Thank you for the wonderful opportunity, my suggestions for further modifications-

1.       Please simplify the title with spelling check (Ref. 3rd line)

2.       Page 3 after line 110- please add the study background with research questions.

3.       Page 4, after line 154-please add the review gaps, novelty and need for the study.

4.       Page4, materials and methods section- please add the following:

a.       Tool validity and reliability

b.      The study limitations

c.       Scope for future research

d.      Data analysis/synthesis plan with software used for the same

e.      Inclusion and exclusion criteria

f.        Methodological justifications for the study period

g.       Measures adapted to remove the study biases

5.       Page 6, line 296- table 1-please add/ specify test value (for example name of the test)

6.       Page 7, line 296- table 1-please add/ specify test value (for example name of the test)

7.       Page 8, line 324- table 2-please add/ specify test value (for example name of the test)

8.       Page 7, after line 423- please add/refine doable suggestions at individual, community and governance levels.  

Thanks and regards,

Author Response

We want to thank the reviewers for helping us to improve our paper.

Author Response

We want to thank the reviewer for the kind open review.

Reviewer 3 Report

Overall an interesting study and presentation. Please clarify if the SCPT questionnaire was validated or not. did you create the questionnaire using the framework cited?

in Materials and Methods you state that the survey was spread in Italian context. Does this mean that the social media was only available in the geographic region? or if it could be seen by those outside Italy?

In conclusion, section second to last paragraph, line 447 the translation to English needs to be corrected.

Author Response

(The authors gave the same response as above.)
